



# 1    Global detection of rainfall triggered landslide clusters

Susanne A. Benz[1,2], Philipp Blum[1]
[1] Karlsruhe Institute of Technology (KIT), Institute of Applied Geosciences (AGW), Karlsruhe, Germany
[2] University of California San Diego (UCSD), School of Global Policy and Strategy (GPS), La Jolla, CA, USA
*Correspondence to*: Philipp Blum (blum@kit.edu) and Susanne A. Benz (sabenz@ucsd.edu)
**Abstract**
An increasing awareness of the cost of landslides on the global economy and of the associated loss
of human life, has led to the development of various global landslide databases. However, these
databases typically report landslide events instead of individual landslides, i.e. a group of landslides
with a common trigger and reported by media, citizens and/or government officials as a single unit.
The latter results in significant cataloging and reporting biases. To counteract this biases, this study
aims to identify clusters of landslide events that were triggered by the same rainfall event. Here the
developed algorithm is applied to the Global Landslide Catalog (GLC) maintained by NASA. The
results show that more than 40 % of all landslide events are connected to at least one other event,
and that 14 % of all studied landslide events are actually part of a landslide cluster consisting of at
least 10 events. However, in a more regional analysis this number ranges from 30 % for the West
Coast of North America to 3 % in the Himalaya Region. The cluster with most landslide events in
a day is located in Rio de Janeiro, Brazil, with 108 events on 6[th] April 2010. In contrast, the longest
running cluster was observed on the West Coast of North America with 132 events occurring in an
area of over 120,000 km$^2$ during 24 days in December 2015. Our study intends to enhance our
understanding of landslide clustering and thus will assist in the development of improved,
internationally streamlined mitigation strategies for rainfall related landslide clusters.



**Keywords:** Landslide events; Database; Extreme weather; Rainfall induced; Early warning systems;

## 1. Introduction

The fatal and catastrophic nature of landslides has led to the development and maintenance of various global databases, such as the NASA Global Landslide Catalogue (GLC; e.g. Kirschbaum et al. 2015) and recently the Global Fatal Landslide Database (GFLD) by Froude & Petley (2018). Typically, these databases have a distinct focus. For example, the Global Landslide Catalogue (GLC) operated by NASA focuses on rainfall triggered landslides (Kirschbaum et al., 2010, 2015), whereas the Global Fatal Landslide Database records fatal landslides (Froude and Petley, 2018; Petley, 2012). Through these databases we are able to provide first estimates on the number of recorded fatalities, which were > 55,000 between 2004 and 2016 (Froude and Petley, 2018) and map near real-time risk for landslides almost on a global scale (Kirschbaum and Stanley, 2018). Still, while they play a key role in understanding the effects of landslides on our society, it is important to note that they are primarily based on news and government reports. These databases therefore do not count landslides, but landslide events, which contain either a single or a multitude of landslides within an area that are assumed to be triggered by the same event (Malamud et al., 2004). The exact number of slope failures in each event is often unknown and depends on the quality of the reporting. For some databases this number is included in a parameter of intensity or size of each event. Typically, for large databases however, this is merely qualitative and describes not only the number of individual landslides, but also an impact such as economic or human losses. This classification is commonly based on press releases and is therefore heavily biased on the news outlet reporting each event (e.g. Carrara et al., 2003).



Landslides triggered by catastrophic events, such as earthquakes or major storms, are often counted
as one event containing thousands of individual landslides (Kirschbaum et al., 2015). In contrast,
landslides caused by non-catastrophic events such as reasonable rainfall, are commonly counted as
individual events, disregarding their shared trigger. Consequently, the overall extent of clustering
in landslides is often unknown. But only if we better understand the extent of clustering between
individual landslide events, will we be able to understand the patterns they occur in and have the
chance to utilize these patterns to improve our forecast models (e.g. Martelloni et al., 2012).
Until now, few studies have focused on rainfall triggered landslide clusters and rather on temporal
clusters over a long time period within a confined region (e.g. Samia et al., 2017; Witt et al., 2010).
Biasutti et al. (2016) investigated the spatiotemporal clustering due to rainfall events for three
selected urban areas of the US West Coast: Seattle, San Francisco and Los Angeles. Over the nine
year study period, they found approximately 20 days within each city with multiple (up to eight)
landslide events. Additionally, they could identify close to 40 landslide events that were followed
by another event within the next week. However, with a focus on only selected study areas, they
did not show the overall extend of these clusters.
The objective of this study is therefore to develop an algorithm, which is able to identify such
clusters on a global scale. By applying the algorithm to the Global Landslide Catalog (GLC) the
overall degree of clustering in the database is shown, and spatial patterns of clusters with at least
10 landslide events are described. Additionally, landslide events and rainfall patterns of the most
intense and longest clusters are comprehensively discussed. In contrast to previous studies, such as
by Biasutti et al. (2016), clusters here are not constricted by a maximum spatial extent, instead they
are grouped by analyzing and comparing rainfall prior to the event at the event locations.



## 2. Material and Method

### 2.1    Landslide Data

All landslide events within this study are part of the Global Landslide Catalog (GLC) operated by

NASA and introduced in Kirschbaum et al. (2010, 2015). Data within the catalogue is based on

online news articles that are found through search engine options such as Google Alerts. In the

presented study, only events with a location accuracy ≤ 25 km are considered. As the rainfall data

used is only available within ± 50° Latitude, landslide events outside of this range are not

considered. Overall, a total of 9279 landslide events, ranging from 1988 to 2018 are analyzed (Fig.

1). However, only 45 of these events occurred before 2007, when the GLC was established.

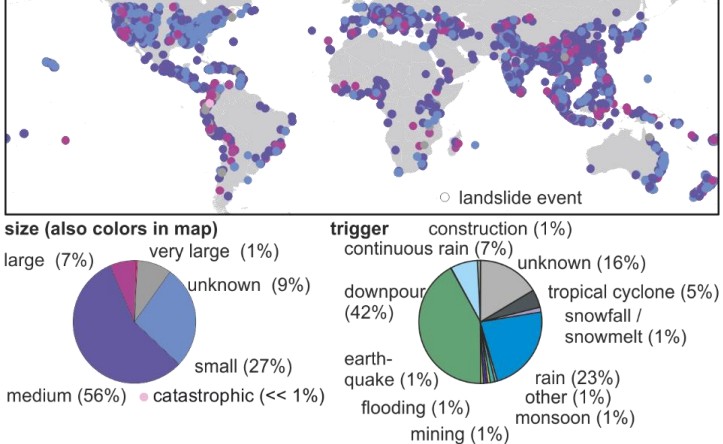

**Figure 1.** Map of all landslide events analyzed in this study and their size (also color in map) and apparent trigger.

Overall a total of 9279 events were tested for clustering.

For each event, the GLC provides a landslide type, e.g. land- or mudslide, and a landslide trigger,

e.g. rainfall, downpour, earthquakes or construction work. Detailed descriptions on these

classifications can be found in Kirschbaum et al. (2010, 2015). Furthermore, within the GLC the

intensity, impact, and number of landslides per event is expressed in a variable called "size". While

events classified as small in the database are only a single landslide, medium or larger landslide





events may consist of multiple landslides within an unspecified range. About 64 % of the studied
events are classified as medium or larger in size. However, a precise count of the number of
landslides contained within these events does not exist in this database nor in any other of the global
scale databases currently available. Within the GLC most of the small events that contain only a
single landslide, are located within the United States (Fig. 1).
**2.2    Rainfall Data**
For the rainfall analysis, the Climate Hazards Group InfraRed Precipitation with Station data
(CHIRPS) (Climate Hazards Group, 2015) is used, which has a resolution of $0.05° \times 0.05°$ and
daily time steps. For each landslide event location, precipitation data were downloaded for 30 days
preceding the event and up to two days after the event using Google Earth Engine (Gorelick et al.,
2017). In order to compare rainfall during the event to overall rainfall at the location, the $95^{th}$
percentile of precipitation excluding non-rainy days was determined for 10 years prior to the event.
This comparison was also previously used by Kirschbaum et al (2015) to identify rainfall triggered
landslide events. However, in their case, rainfall data from the Tropical Rainfall Measuring Mission
(TRMM) was used for the time period 2000–2013 independent of the date of the landslide event.
Due to the higher spatial resolution CHIRPS data was used here instead.
In addition to the $95^{th}$ percentile of rainfall, the global rainfall threshold by Guzzetti et al. (2008)
was also utilized to determine the likelihood of the individual landslide events being triggered by
rainfall. In their study 2626 rainfall events that have resulted in shallow landslides and debris flows
were analyzed in order to determine the following global rainfall intensity–duration threshold
[http://rainfallthresholds.irpi.cnr.it]:

$$I = 2.2 \cdot D^{-0.44} \tag{1}$$



Here the threshold intensity ($I$) was determined for each 24 hours starting with a duration ($D$) of
12 hours. This results in an average precipitation of 0.73 mm/h for $D = 12$ h, 0.45 mm/h for $D =$
36 h, and 0.35 mm/h for $D = 60$ h. The rainfall threshold was then compared to the cumulative
mean precipitation of the rainfall event preceding each landslide event.
**2.3    Method**
The main objective of this study is to identify clusters of landslide events that occurred during, and
are likely triggered by the same rainfall event. To determine if two events, A and B, occurred during
the same rainfall event, two conditions have to be fulfilled: (I) A and B occurred within three days
of each other, and (II) spearman correlation between daily precipitation at A and at B is > 0.7 and
has a p-value < 0.05 for the 30 days preceding the later of the two events. Other landslide events
that fulfill these conditions with either A or B, are considered to be part of the cluster. A schematic
drawing of this algorithm is given in Fig. 2. The threshold value of three days maximum between
two events was used following Biasutti et al. (2016), who found it unlikely that landslide events
occurring more than three days apart, occurred during the same rainfall event. However, it is
important to note that their study was set in three metropolitan areas on the West Coast of the USA
and might not be applicable everywhere. The threshold value of the spearman correlation
coefficient was determined by testing the robustness of the identified clusters for different threshold
values between zero and one (Fig. S1). It was set to be 0.7 as from here on numbers of landslides
per cluster, duration of clusters, and area of clusters are stable for their mean and maximum values
(Fig. S1).





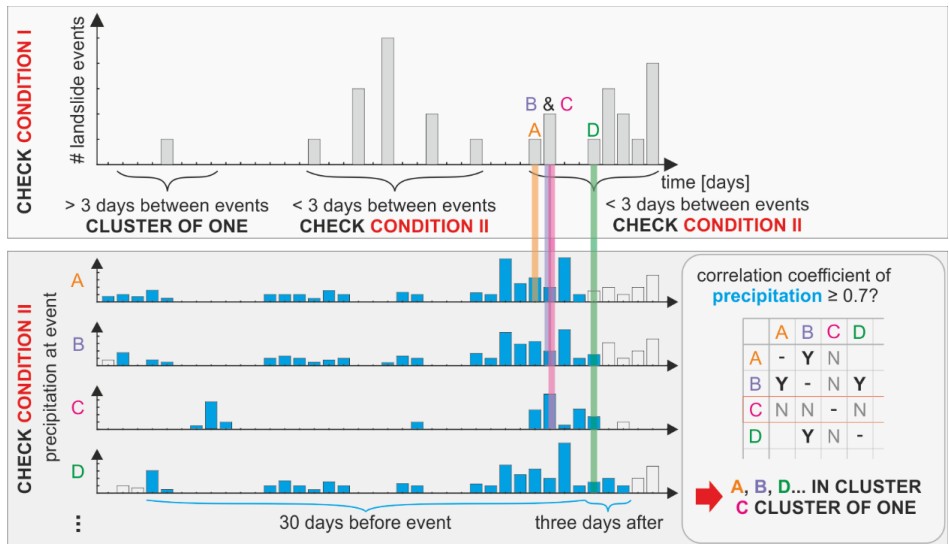


**Figure 2.** Schematic drawing of the algorithm used to identify, if two landslide events within the Global Landslide

Catalog (GLC) occurred during the same rainfall event and hence belong to the same cluster. For condition II only

events occurring within three days of each other are compared.

## 3.    Results and Discussion

### 3.1    Clustering Characteristics

The presented algorithm divided the 9279 landslide events of the Global Landslide Catalog (GLC)

into 6474 clusters of events connected through precipitation. However, 85 % of these clusters

consist of only a single landslide event, containing in total 59 % of all recorded landslide events.

This implies that a large number of landslide events are in fact isolated events with no association

to other events. Nevertheless, 67 % of these 'single landslide event'-clusters are categorized as

medium or larger and might contain more than one landslide (in comparison 58 % of the landslide

events in clusters ≥ one landslide event are categorized as medium or larger). Hence, the number

of isolated landslides is likely to be significantly smaller than the number of isolated landslide

events.



In the Global Landslide Catalog (GLC) only 3 % of the analyzed landslide events are linked to
triggers unrelated to rainfall such as construction, volcanos or earthquakes. This number is reduced
to 1.5 % for landslides in a cluster of more than one event. Due to the low number of events in this
category, future research is necessary to test and thoroughly validate these findings as well as to
assess possible reasons and implications of this phenomenon. For now, we assume that this is
mainly caused by biased reporting and cataloging of landslide events, where events linked to larger
disasters such as earthquakes, might be reported as one large landslide event, whereas landslides
linked to rainfall, might be individually reported. Similar observations were previously made by
Kirschbaum et al. (2015) for events in the GLC that are linked to major storms. An example of this
is the catastrophic magnitude 7.8 Gorkha earthquake in Nepal in 2015. While more than 25,000
landslides occurred during the earthquake and its aftershock sequence (e.g. Roback et al., 2018),
they are only reported as 13 landslide events in the excerpt from the GLC analyzed here. In it, they
are described as ranging in size from small to large and their trigger is given as "unknown",
"earthquake" and in one case "snowmelt". Our algorithm sorts these events into eight clusters of
up to three events.



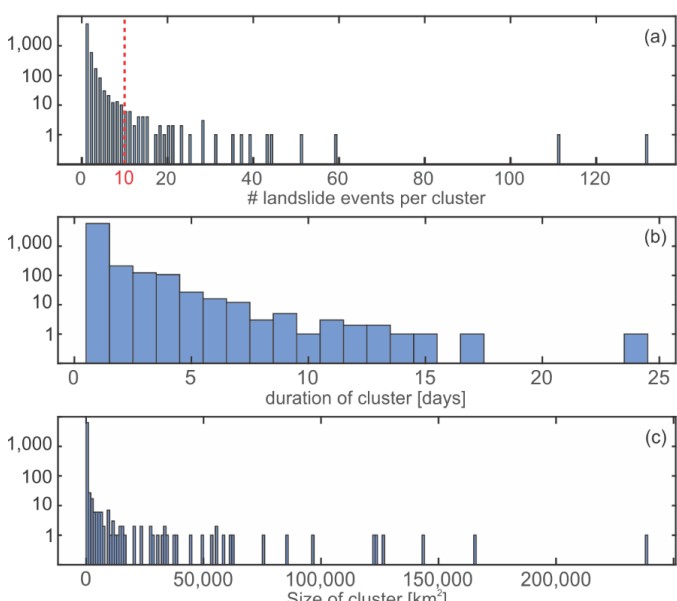


**Figure 3.** Histogram of the number of events per cluster, duration of clusters and area of the convex hull of each cluster.

Clusters with only a single landslide event were appointed an area of zero. Within this study, all clusters with at least
10 landslide events were analyzed more closely.
Figure 3 provides histograms of the landslide events per cluster, duration of clusters and area
covered by clusters (convex hull) in a logarithmic scale. As expected, for all three aspects frequency
reduces drastically for larger numbers. In the following section all 50 clusters with at least 10 events
(marked in red in Figure 3) are evaluated more closely.



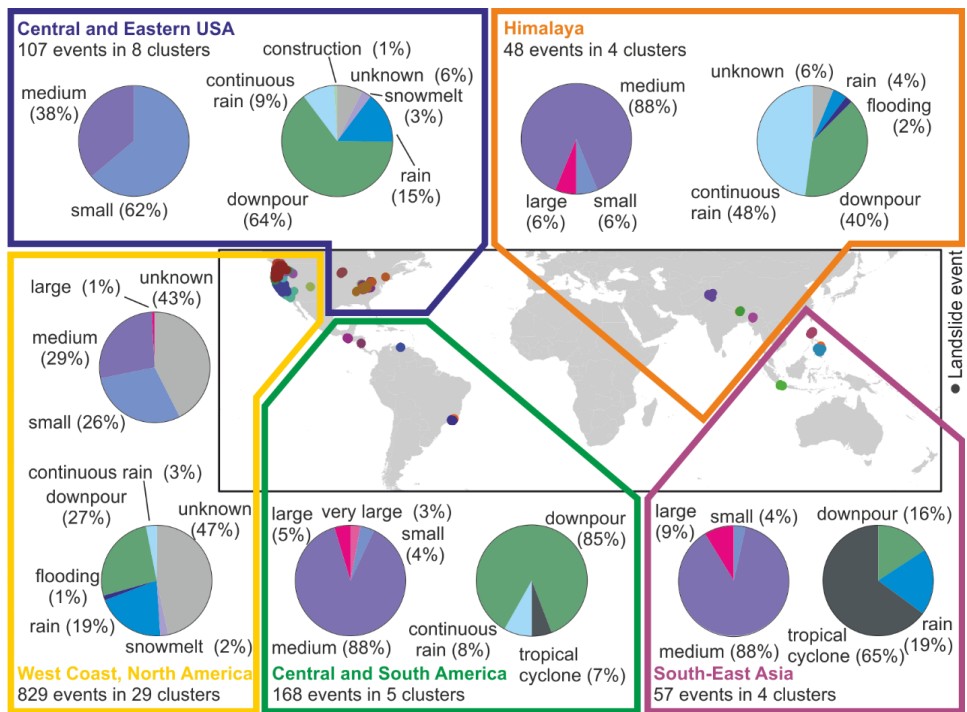


**Figure 4.** Location of all landslide events within clusters ≥ 10 events (different colors indicate different clusters).

Overall, clusters in five distinct regions could be identified in the GLC (see Table S1 for more detail). Size and trigger
(GLC categorization) of the associated landslide events are also shown.

## 3.2    Clusters with more than ten Landslide Events

### 3.2.1    Global Analysis

Table S1 gives more detail of the 50 clusters with at least 10 events. In total 13 % of all landslide
events are associated with one of these clusters. As the database is most likely incomplete, the true
number is expected to be higher. Overall the algorithm detects clusters in five distinct regions: (1)
West Coast of North America, (2) Central and Eastern USA, (3) Central and Southern America, (4)
Himalaya Region and (5) South-East Asia (Fig. 4). However, close to three quarters of all clusters
≥ 10 events are found within the USA mostly due to a bias in the GLC database (Kirschbaum et
al., 2015) (Fig. 1). This is also shown in the size of recorded landslide events (Fig. 4).



In North America events are often classified as small in size, while clusters in the other regions
contain mainly medium events. This might be due to English speaking media, on which the GLC
is based, only picking up on large international events that consist of multiple landslides within an
area and smaller ones are under or not reported at all.
The median clusters with at least 10 events last six days, consist of 15 events, and span over an
area of 15,000 km² (Fig. 5). As expected, there is a positive correlation between cluster duration
and area (spearman correlation coefficient of 0.70, p-value: 0.001). However, this cannot be
observed for cluster duration and number of landslide events within the cluster (spearman
correlation coefficient of 0.44, p-value: 0.001). When comparing the different regions, clusters
located on the West Coast of North America are on average the longest and cover the largest area.
In contrast, events in South America are shortest and smallest, nevertheless they have the highest
number of events and clusters per day (Table 1).
On a global scale, no significant trend over time can be observed and clusters with ≥ 10 events
occur around the year (Fig. S2). Similarly, the total number of reported landslide shows no
significant increase in the GLC (Kirschbaum et al., 2015) as well as in other global databases such
as the Global Fatal Landslide Database (Froude and Petley, 2018). More regional observations
show seasonal variation and are described more closely in the following chapters. However, for
three out of the five regions, there are only five clusters or even less.





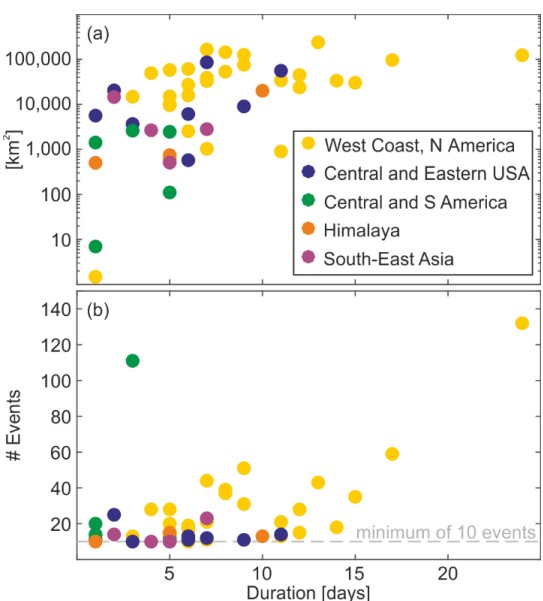


**Figure 5.** Link between the duration of the individual clusters ≥ 10 events and a) the covered area and b) the number

of landslide events per cluster. The color of the scatter plots indicates the region, in which each cluster occurred.

**Table 1.** Regional statistics for all landslide clusters (LC) with at least ten landslide events (LE).

| Region | # LC | # LE | LE per LC | Average duration of LCs | LEs per day | Average area of LCs [km²] | Percentage of LE in a LC ≥ 10 LE |
|---|---|---|---|---|---|---|---|
| West Coast, North America | 29 | 829 | 28.6 | 9 | 3.3 | 52,970 | 31 |
| Central and Eastern USA | 8 | 107 | 13.4 | 6 | 2.4 | 23,357 | 12 |
| South and Central America | 5 | 168 | 33.6 | 3 | 11.2 | 1,320 | 18 |
| Himalaya | 4 | 48 | 12.0 | 5 | 2.3 | 5,476 | 3 |
| South-East Asia | 4 | 57 | 14.3 | 5 | 3.2 | 5,143 | 4 |


### 3.2.2 West Coast, North America

Landslides in the west of North America have been intensively investigated, mainly in the form of

case studies that discuss landslides along the Pacific coast in the states of California (Collins and

Sitar, 2008; Wieczorek, 1988), Oregon (Benda, 1990; Miller and Burnett, 2008) and Washington





(LaHusen et al., 2016; Perkins et al., 2017). This region is also one of the few, where the clustering
of rainfall triggered landslide events was previously investigated, showing qualitatively that there
are many instances in which landslides occur on consecutive days (Biasutti et al., 2016).
About 31 % of all landslide events recorded in this area belong to a cluster of at least ten events.
This is the highest number compared to the other regions of the world (Table 1). However, this
effect might be amplified by the high number of reported landslides. The large number of events
and clusters is mainly due to geologic, topographic, climatic conditions and construction practices.
For example in Oregon, steep slopes and heavy rainfalls are as well as poor construction practices
result in high economic losses (Wang et al., 2002). Burns et al. (2017) estimated an average annual
loss of $15.4 million due to landslides in Oregon alone. In years with heavy storms such as 1996,
this can accumulate to more than $100 million (Wang et al., 2002).
The observed clusters in this area are among the longest and have the largest areas of all regions
(Table 1). While the size of landslide events (as given by the GLC) in the west of North America
are small compared to most other regions, there is also a considerable amount of events, where the
size is unknown (43 %, Fig. 4). While about half of the landslide events within clusters $\geq 10$ events
are classified as "trigger unknown" (47 %), landslide events with a known cause are mainly
triggered by downpour (27 %) or rain (19 %) (Fig. 4). However, when looking at satellite based
rainfall data preceding the clusters, rainfall cannot always be identified as a trigger (Fig. S3). While
it generally exceeds the global rainfall threshold (Guzzetti et al., 2008), the 95[th] percentile of
precipitation on rainy days is not reached for the majority of the clusters. Although, several studies
linked landslides within California to earthquakes (e.g. Harp and Jibson, 1996; Keefer, 2000), they
occurred before 2007 and are not registered in the GLC.



While there appears to be no significant change in the number of clusters over time (Fig. S2), most
clusters occur during the rainy season (November to March), when most landslide events occur.
Within the west of North America this time period is therefore often referred to as the "landslide
season" (e.g. Mirus et al., 2018). Only one cluster in this region appears in June (Cluster ID 21,
Table S1). However, the center of this cluster is located more inland (in San Miguel County,
Colorado) and is also the shortest cluster (only one day) within the region as well as the most local
of all clusters in this study, covering only 1 km$^2$. While this cluster is triggered by downpour
according to the GLC, this is not apparent from satellite derived precipitation (Fig. S3). The small
size of the cluster might be the reason, why low-resolution satellite derived precipitation does not
record any anomalies here.

### 3.2.3    Central and Eastern USA

While most of the clusters with ≥ 10 landslides events of this region, are located in the Appalachian
Plateau (Ohio, West Virginia and Kentucky), one cluster can be found in Minnesota (ID 34 in Table
S1 and Fig. S4). While it is considerably smaller (580 km$^2$ compared to > 9,000 km$^2$), it is
comparable to the Appalachians cluster in its number of landslide events and duration. The
Appalachian Plateau is well known for its landslides and the annual direct cost in Kentucky exceeds
$10 million (Crawford and Bryson, 2017).
Like the landslide clusters observed in the west of North America, clusters here consist mainly of
small landslides, which is most likely linked to the news alerts on which the GLC is based.
Checking sources in the GLC, they are mainly reported within smaller, more local news outlets
compared to landslide events outside of the US. To our knowledge the individual events grouped
by our algorithm into clusters have never been linked before. Clusters in this region occur
predominantly in spring (February to June), when rainfall is highest, slightly later than events on



the West Coast (Fig. S2). According to GLC they are predominantly triggered by downpours (64 %,
Fig. 4). However, extreme rainfall is not always visible in satellite derived precipitation (Fig. S4).
For most clusters, it is below the 95$^{th}$ percentile, but above the global threshold. It is worth noting
than one cluster located in West Virginia (Cluster ID 35) shows no rainfall on the satellite before
day three of the cluster. Following the GLC, early landslide events within this cluster are linked to
snowmelt.
**3.2.4   Central and South America**
In contrast to the clusters in North America, more than 95 % of landslide events within clusters of
this region are medium in size or larger and might consists of several landslides themselves (Fig.
4). Thus, the number of landslides per cluster and per day is likely to be significantly higher than
the number of events per cluster and per day. Still, clusters in this area are on average only two and
a half days long, covering an area of slightly over 1,500 km² and they are the smallest and shortest
of all regions (Fig. 5, Table 1). It is important to note that this region covers the largest area reaching
from Rio de Janeiro in Brazil to Guatemala in Central America. From the few clusters we could
identify, it appears that there are dissimilarities between the clusters in Central America and South
America. The two clusters in Nicaragua (ID 42) and Guatemala (ID 39) are triggered by continuous
rain and a tropical cyclone, respectively. In contrast, all events located in South America (IDs 38,
40, and 41) are all triggered by downpour (Table S1 and Fig. S5).
**3.2.5   Himalaya**
Like in South America, most landslide events (94 %) associated with clusters with ≥ 10 events in
the Himalaya region are categorized as medium and larger. Thus, the number of landslides per
cluster is again expected to be significantly higher than the number of landslide events per cluster.
However, there may be differences between regions. Event ID 44, located in India and Pakistan



around Jammu and Kashmir, is classified as medium to small, much longer (10 days) and covers
an area more than 10 times larger than the other clusters. All of them are classified as medium or
large and are located in the East of India with some events in Nepal (Table S1). In both regions,
clusters are triggered by continuous rain or downpour. For all clusters satellite based rainfall data
exceeds the global threshold, and in most cases the 95th percentile of rainfall on rainy days (Fig.
S6). It is important to note that while earthquake triggered landslides are common in the region
(e.g. Parkash, 2013; Roback et al., 2018), the presented algorithm is by design only able to pick up
clusters that are linked by rainfall.
**3.2.6 South-East Asia**
As only four clusters are identified in this region, a detailed analysis is impossible. Again, 96 % of
the events associated are categorized as medium or larger and the main triggers are tropical
cyclones (Cluster IDs 47 and 48), downpour (Cluster ID 49), and rain (ID 50) (Table S1). Here,
satellite based rainfall data before clusters is both above the global rainfall threshold and in most
cases above the $95^{th}$ percentile (Fig. S7). While only one of the four clusters (ID 50) is recorded
outside of the Philippines (in Indonesia), there is no apparent difference between both countries
(Table 1).
**3.3 Most Intense Cluster**
The cluster with the most events in one day, i.e. most intense cluster, happened in Rio de Janeiro,
Brazil, as well as neighboring cities Niteroi and Sao Goncalo in 2010. In an area of approximately
2,800 km$^2$, 111 landslide events were recorded within only three days, however predominantly on
6$^{th}$ April 2010 (Table S1, ID 38). This is almost four times as many landslide events in a single day
than the second most intense clusters (IDs 1 and 3) located in Washington and Oregon, USA. Both
recorded 29 events in one day.

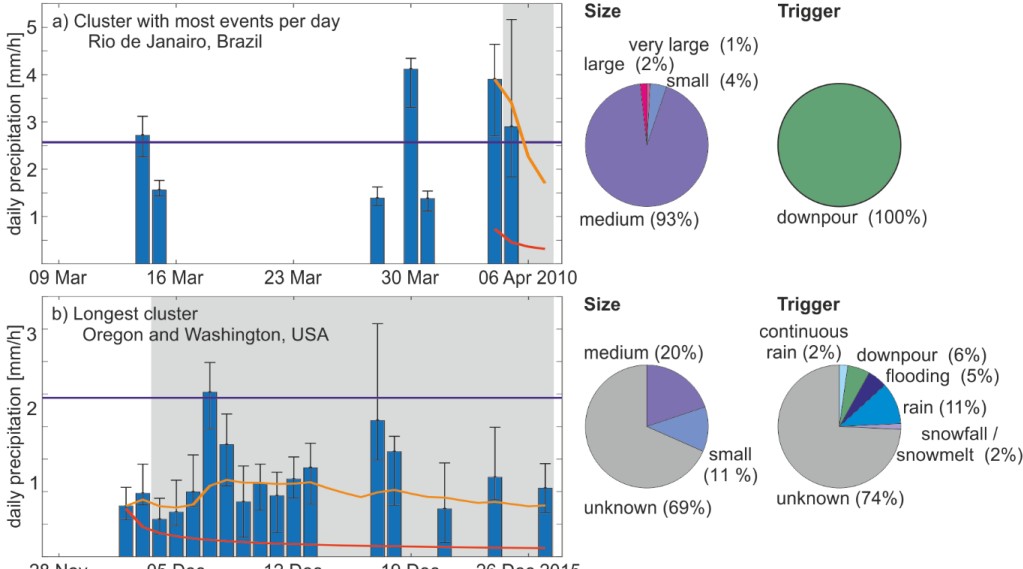


**Figure 6.** Daily precipitation for 30 days preceding the last landslide event of the cluster with the size of the associated

landslide events and their trigger according to the GLC. Shown is the median precipitation for all landslide locations

with the inner quartiles as an error bar. The 95th percentile of daily rainfall (rainy days only) in the ten years preceding

the event is given in blue, in red the global rainfall threshold ID (Guzzetti et al., 2008) and in orange the cumulative

mean for the rainfall event preceding the cluster. a) Cluster with the most events per day (ID 43), and b) longest running

cluster (ID 22).

Most of the 111 events associated with the cluster in Rio de Janeiro were recorded as medium in
size, all of which were triggered by downpour (Fig. 6a). This is confirmed by satellite derived
precipitation. Heavy rainfalls (Figs. 6a, 7) occurred on the 4th and 5th of April of up to 210 mm per
day. In comparison, the 95th percentile in the 10 years preceding this cluster is on average only 62
mm per day (rainfall for each individual location shown in Fig. S8). While the rainfall covered a
large area, landslide events were primarily reported for steep slopes just outside the densely
populated city center. Due to its location close to, and inside the urban area of Rio de Janeiro, the
cluster caused approximately 200 fatalities according to CNN news reports
(http://www.cnn.com/2010/WORLD/americas/04/12/brazil.flooding.mudslides/).



The location in the city might also be the reason for the large number of events being reported, as
we can expect more individual landslides being reported here compared to the countryside.
While studies not based on English speaking news alerts report a large number of landslides within
and around Rio de Janeiro (Calvello et al., 2015; Sandholz et al., 2018), only nine additional
landslide events inside the area of this cluster were reported in the GLC between 2009 and 2018.
Additionally, just northwest of the cluster another cluster occurred in January 2011 (ID 41 in Table
S1, Fig. S5). Although, this cluster only counts 20 individual landslide events within the GLC, it
is being reported as thousands individual landslides (Coelho Netto et al., 2013).

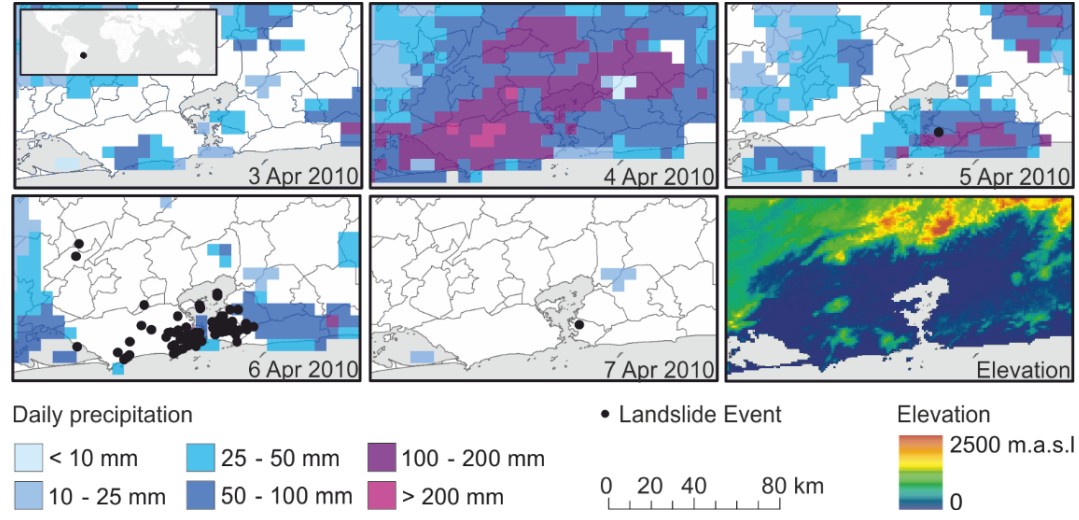

**Figure 7.** Location of the events in the cluster with the most events per day located in Rio de Janeiro, Brazil. Also
shown are daily precipitation and elevation. Elevation data is taken from the US Geological Survey (GTOPO30).
**3.4    Longest Cluster**
The longest running cluster identified in this study occurred in Oregon and Washington, USA from
4th to 27th December 2015 for a total of 24 days with 132 landslide events (Cluster ID 18, Table
S1). The second longest cluster lasted 17 days over January and February in 2012 and was also





located in Oregon and Washington, USA (Cluster ID 7). Overall, most events within the longest
cluster are unknown in size (69 %) and trigger (74 %) (Fig. 6b). However, inspecting satellite based
rainfall data, continuous rainfall appears to be the main trigger (Fig. 6b, Fig. 8 and Fig. S9 for
rainfall at the individual event locations). While daily rainfall is mainly below the 95th percentile,
cumulative mean rainfall is continuously above the global rainfall threshold. Although, heavy
rainfall is common in this area during winter times, for this cluster it lasted longer than usual and
was followed by shorter rain events in short successions (Fig. 8). Thus, the series of landslides did
not halt resulting in the longest cluster in the GLC. Following the information on sources within
the GLC, it appears that local media reported about the individual landslide events, but did not
detect on the extreme length of the continuous series of landslide events at this point in time (e.g.
https://kval.com/news/local/landslide-blocks-i-5-in-sw-washington;
https://q13fox.com/2015/12/09/landslide-above-puget-sound-damages-several-homes-at-least-
one-vehicle/). As landslide events are such a common occurrence in this region, and due to the
large area covered by this cluster, there is currently little to no emphasis on the longevity of this
specific series of landslide events in media and scientific studies.



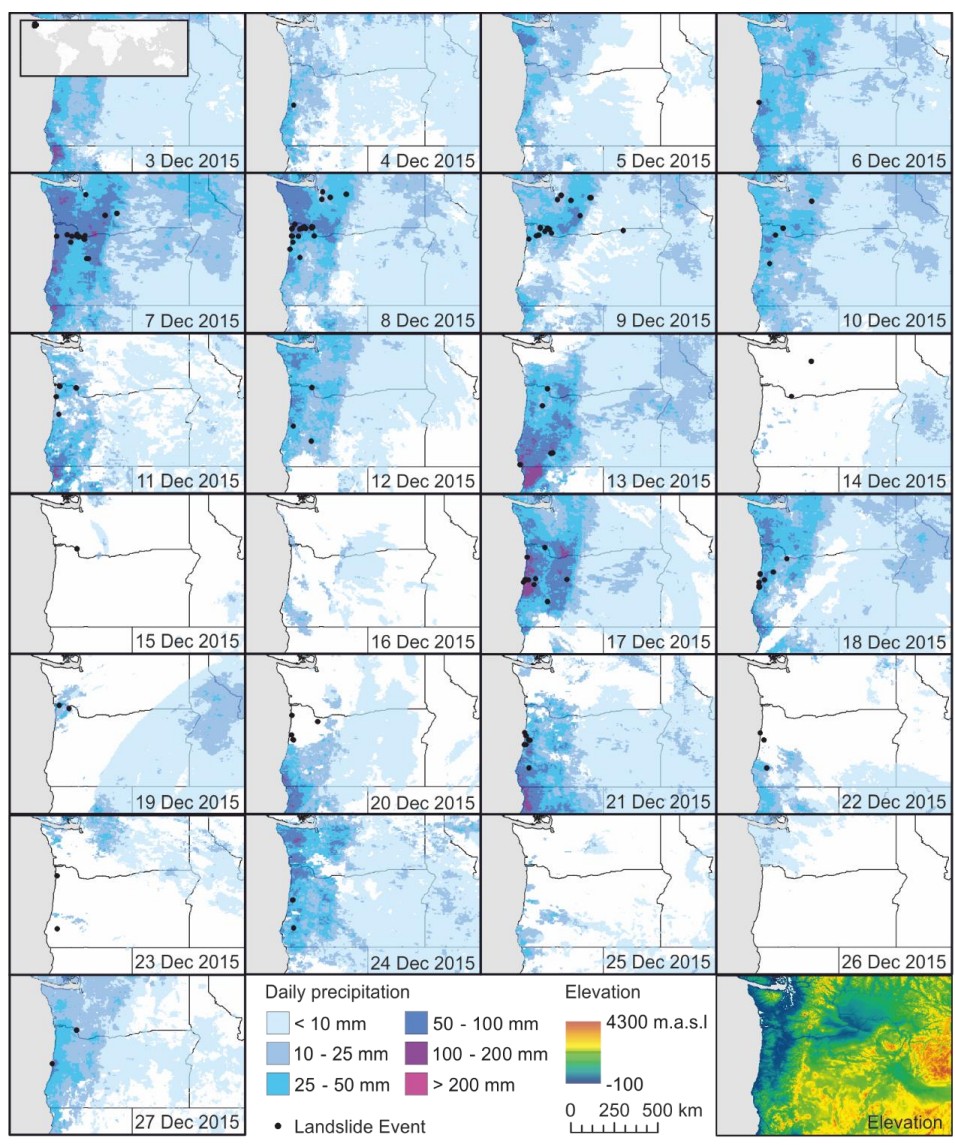

**Figure 8.** Location and time series of the longest cluster, located mainly in Oregon, USA. Also shown are daily rainfall and elevation. Elevation data is available from the US Geological Survey (GTOPO30).

## 4. Conclusion

In this study an algorithm is presented that detects clusters of landslide events that occur during, and are likely triggered by the same rainfall events. Here this algorithm is applied to the Global





Landslide Catalog (GLC), where it detects that more than 40 % of all recorded events can be linked
to at least one other event. The global analysis shows that 14 % of all landslide events are part of a
cluster ≥ 10 events. However, this percentage varies dramatically by the region, ranging from 30 %
on the West Coast of North America to 3 % in the Himalayas. Part of this is caused by sampling
and reporting bias. As the GLC is based on English speaking media, events in the USA are reported
and cataloged in much greater detail than events abroad. Nevertheless, within the GLC we could
detect clusters ≥ 10 landslide events in five distinct regions: (1) West Coast of North America, (2)
Central and Eastern USA, (3) Central and Southern America, (4) Himalaya Region, and (5) South-
East Asia. In South America, the studied clusters are the shortest, but contain the most events per
day. However, this is mainly due to a cluster in Rio de Janeiro, where 108 of events were recorded
on $6^{th}$ April 2010. As most of these events are classified as medium or larger, the absolute number
of landslides is expected to be significantly higher. In contrast, the longest and largest clusters are
observed on the West Coast of North America. On average clusters here last nine days and cover
an area of more than 50,000 $km^2$. The steep slopes and continuous rainfalls present in the area
combined with the above average reporting of landslide events, makes a more detailed analysis of
rainfall related landslide clusters possible. The longest of all detected clusters ≥ 10 landslide events
is also located in this region: In December 2015, 132 landslide events were recorded over a time
period of 24 days spanning more than 120 thousand $km^2$, which were all triggered by the same
rainfall event. Detection of large scale clusters such as this one can not only help to improve our
understanding of the link between individual events, but also be used in our mitigation strategies.
Only once we improve our understanding of the relation between individual landslide events, we
will be able to predict their behavior and forecast their economic losses and fatalities. While our
study does not replace case specific and small scale studies, as well as the identification of threshold
values, it can provide an improved understanding for managing landslide mitigations on a larger





scale. Within the area covered by individual clusters the same mitigation strategies, including early
warning systems (EWS) based on weather forecast simulations, can be developed and validated.
For future research we recommend to use the presented algorithm not only for the correlation with
precipitation data, but also to include the geometry of atmospheric rivers during cluster detection.
Finally, the algorithm could be applied to more regional and other global landslide databases
thereby improving our understanding on the spatial and temporal occurrence of landslide clusters.

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
