# Peer review of "Global detection of rainfall triggered landslide clusters"

_Natural Hazards and Earth System Sciences, 2018_

## Referee Comment (RC1) · Anonymous Referee #1 · 26 Feb 2019

The submitted paper deals with the identification of clusters in the Global Landslide Catalogue (GLC) in relation to individual rainfall events extracted from CHIRPS precipitation data. It is therefore also related to the investigation of biasing effects in small-scale (global) landslide catalogues that are not compiled through landslides inventorying but rather based on media or governmental reports. In this respect, the paper is timely, interesting and well suited to NHESS. The presentation is clearly structured and the language of the article is fluent, the Figures are of good quality. The clustering algorithm proposed by the authors to relate reported landslides to individual rainfall events is interesting and may be worth to be published. However, the validity of the constraints of the clustering algorithm is not discussed in detail by the authors. In this respect, a sensitivity analysis is missing in such that different time span thresholds in precipitation

may be tested in order to investigate the behavior of the identified clusters. Also, the effect of different values for the spearman coefficient to discriminate clusters may be investigated. Another issue is that the only constraint applied for landslide clustering is precipitation. Environmental information like e.g. climatic setting, subsoil lithology or relief parameters are not introduced into the clustering algorithm and the effect of introducing those is not investigated or discussed.

---

## Referee Comment (RC2) · Anonymous Referee #2 · 17 Apr 2019

General comments The paper proposes an algorithm to detect and group clusters of landslide events that occurred or were triggered by the same rainfall event. The algorithm is then applied to the Global Landslide Catalogue (GLC). The paper has a good structure, even if, some improvements are needed to increase its quality and clarity. The research topic, from my point of view, is useful and of interest. In the following my revision. The abstract should contain a description of the main aim and the innovative features of the research proposed. In my opinion, the abstract focuses too much on the results. The rows 14-20 should be summarized. It is not useful, and probably counterproductive, going too much into detail in the abstract. Then, I would suggest the authors to describe the aim of the algorithm and the innovative features of the research, focusing on how the paper is pushing a step forward in this topic. Concerning

section 2.3 "Method", I would suggest going more deeply into the explanation of the algorithm. A flowchart can be useful to fully describe the processes behind it. Moreover, the description of the two conditions for gathering landslide events into the same rainfall events should be described more in detail, in particular the condition (II). The choice of those values, for the spearman correlation and the p-value, should be fully described, also commenting about the limitations connected to those choices (as done for condition (I)). Then, I would also suggest moving in this section the method used to define the rainfall events, using section 2.1, and 2.2 only to describe the dataset available. Please consider using more tables in Section 3 to summarize and better describe the results obtained. Currently the text may result a bit confusing. Finally, I would suggest to clearly split the discussion from the results. Please consider creating Section 4. "Discussion. In this way the authors' comments are highlighted and easy to be understood for a reader.

Specific comments Avoid in the text: ">"," <", etc... Fig.1 difficult to distinguish among dots. I would suggest enlarging this figure. Please check the numbers for tables and figures (i.e., lines 122, 124: Fig. S1 ??; line 169: Table S1??)

---

## Author Comment (AC1) · 29 May 2019

**Response to Reviewers of the manuscript**

**"Global detection of rainfall triggered landslide clusters"**

**by Benz and Blum, submitted to *Natural Hazards and Earth System Sciences*.**

Manuscript Number: nhess-2018-391

Revision due before: 29 May 2019

**Reviewer #1 comments:**

**Comment I:**
The submitted paper deals with the identification of clusters in the Global Landslide Catalogue (GLC) in relation to individual rainfall events extracted from CHIRPS precipitation data. It is therefore also related to the investigation of biasing effects in small-scale (global) landslide catalogues that are not compiled through landslides inventorying but rather based on media or governmental reports. In this respect, the paper is timely, interesting and well suited to NHESS. The presentation is clearly structured and the language of the article is fluent, the Figures are of good quality. The clustering algorithm proposed by the authors to relate reported landslides to individual rainfall events is interesting and may be worth to be published. However, the validity of the constraints of the clustering algorithm is not discussed in detail by the authors. In this respect, a sensitivity analysis is missing in such that different time span thresholds in precipitation may be tested in order to investigate the behavior of the identified clusters. Also, the effect of different values for the spearman coefficient to discriminate clusters may be investigated.

**Reply:** Thank you very much for the kind words and constructive comments.
We agree and therefore included a sensitivity analysis, which investigates the different time span thresholds in precipitation and the effects of different spearman coefficients. The following discussion is now added:

*"The threshold value of the spearman correlation coefficient was determined by testing the robustness of the identified clusters for different threshold values between zero and one (Fig. S2). Our results indicate that mean duration, area, and number of landslides per cluster are comparably robust to changes of the spearman correlation coefficient. In contrast maximum duration, area and number of landslides per cluster change drastically for different threshold values. From a correlation coefficient threshold of 0.35 to 0.7, maximum number of landslide events per cluster decreases from close to 500 to slightly above 100, maximum duration decreases from more than 80 days to approximately 25, and area decreases from 60,000,000 km² (approximately 1/3 of the planet's surface area) to 200,000 km². For threshold values greater 0.7, only minor changes are observed. Hence, the latter was set as the correlation threshold value for this study (Fig. S2).*

*Additionally, we tested the robustness of the method to the time period of precipitation for which the correlation coefficient was determined (Fig. S3). It appears that the number of days is much less influential than the set correlation coefficient threshold (Fig. S2). Again, maximum number of landslides, area, and duration are impacted most, however remain stable for time period longer than 30 days prior to the second event.''*

[Figure]

***Figure S3.*** *Impact of the chosen threshold for the timespan for which the spearman correlation coefficient is determined on the total number of clusters in the global landslide catalog, on the average number of landslides per cluster, on the average duration of landslide clusters, and on the average area of landslide clusters. The correlation coefficient threshold was set to 0.7 for this analysis. In this study a threshold of 30 days was chosen, as from this point onwards number of clusters, maximums size of, duration of and landslides per cluster, becomes stable.*

**Comment II:**
Another issue is that the only constraint applied for landslide clustering is precipitation. Environmental information like e.g. climatic setting, subsoil lithology or relief parameters are

not introduced into the clustering algorithm and the effect of introducing those is not investigated or discussed.

**Reply:** Correct, the introduced algorithm focus solely on rainfall, detecting clusters of landslides triggered by the same rainfall event. While different lithology and relief parameters impact the rainfall intensity-duration threshold, two landslide events located in different areas with different thresholds might still be triggered by the same rainfall event. We therefore decided to not include any of these parameters in the algorithm. However, an additional sentence was added to the chapter describing the algorithm that discusses this issue:

"*The introduced algorithm is independent of subsoil topography and relief parameters. While these impact the precipitation intensity-duration threshold that is commonly expected to trigger landslides, locations with different thresholds might still experience landslides triggered by the same rainfall event.*"

---

## Author Comment (AC2) · 29 May 2019

**Response to Reviewers of the manuscript**

**"Global detection of rainfall triggered landslide clusters"**

**by Benz and Blum, submitted to *Natural Hazards and Earth System Sciences.***

Manuscript Number: nhess-2018-391

Revision due before: 29 May 2019

**Reviewer #2 comments:**

**Comment I:**
The paper proposes an algorithm to detect and group clusters of landslide events that occurred or were triggered by the same rainfall event. The algorithm is then applied to the Global Landslide Catalogue (GLC). The paper has a good structure, even if, some improvements are needed to increase its quality and clarity. The research topic, from my point of view, is useful and of interest. In the following my revision.

The abstract should contain a description of the main aim and the innovative features of the research proposed. In my opinion, the abstract focuses too much on the results. The rows 14-20 should be summarized. It is not useful, and probably counterproductive, going too much into detail in the abstract. Then, I would suggest the authors to describe the aim of the algorithm and the innovative features of the re-search, focusing on how the paper is pushing a step forward in this topic.

**Reply:** We agree. Hence the abstract was rewritten accordingly:

> *"An increasing awareness of the cost of landslides on the global economy and of the associated loss of human life, has led to the development of various global landslide databases. However, these databases typically report landslide events instead of individual landslides, i.e. a group of landslides with a common trigger and reported by media, citizens and/or government officials as a single unit. The latter results in significant cataloging and reporting biases. To counteract this biases, this study aims to identify clusters of landslide events that were triggered by the same rainfall event. An algorithm is developed that finds a series of landslide events that a) is continuous with no more than two days between individual events, and b) precipitation at the location of an individual event correlates with precipitation of at least one other event. The developed algorithm is applied to the Global Landslide Catalog (GLC) maintained by NASA. The results show that more than 40 % of all landslide events are connected to at least one other event, and that 14 % of all studied landslide events are actually part of a landslide cluster consisting of at least 10 events and up to 108 events in one day. Duration of the detected clusters also varies greatly from 1 to 24 days. Our study intends to enhance our understanding of landslide clustering and thus will assist in the development of improved, internationally streamlined mitigation strategies for rainfall related landslide clusters."*

**Comment II:**

Concerning section 2.3 "Method", I would suggest going more deeply into the explanation of the algorithm. A flowchart can be useful to fully describe the processes behind it.

**Reply:** We agree. Hence, an additional and more detailed flowchart was created. As Fig. 2 already provides a simplified schematic drawing, we decided to place the flowchart in the supplementary material.

[Figure]

*Figure S1. Flowchart of the algorithm to detect clusters within the global landslide catalog (GLC). Symbols included: ∀ - for all; ∧ - logical conjunction; ∈ - element of; ∄ - there does not exist; ∋ - contains.*

**Comment III:**

More-over, the description of the two conditions for gathering landslide events into the same rainfall events should be described more in detail, in particular the condition (II). The choice of those values, for the spearman correlation and the p-value, should be fully described, also commenting about the limitations connected to those choices (as done for condition (I)).

**Reply:** We agree. Hence, following additional feedback from reviewer I, the sensitivity analysis was severely extended and the choices set for condition (II) are discussed in more detail. Additionally, a more inclusive discussion of limitations is now included:

*"The threshold value of the spearman correlation coefficient was determined by testing the robustness of the identified clusters for different threshold values between zero and one (Fig. S2). Our results indicate that mean duration, area, and number of landslides per cluster are comparably robust to changes of the spearman correlation coefficient. In contrast maximum duration, area and number of landslides per cluster change drastically for different threshold values. From a correlation coefficient*

*threshold of 0.35 to 0.7, maximum number of landslide events per cluster decreases from close to 500 to slightly above 100, maximum duration decreases from more than 80 days to approximately 25, and area decreases from 60,000,000 km² (approximately 1/3 of the planet's surface area) to 200,000 km². For threshold values greater 0.7, only minor changes are observed. Hence, the latter was set as the correlation threshold value for this study (Fig. S2).*

*Additionally, we tested the robustness of the method to the time period of precipitation for which the correlation coefficient was determined (Fig. S3). It appears that the number of days is much less influential than the set correlation coefficient threshold (Fig. S2). Again, maximum number of landslides, area, and duration are impacted most, however remain stable for time period longer than 30 days prior to the second event.*

*It is important to note that the introduced method does not limit the spatial extent of the found landslide clusters. While this ensures that previously undetected, large-scale connections between individual landslide events are found, it is also susceptible to link landslides occurring in different parts of the world, where rainfall coincidentally correlates. Hence, when applying the method to another dataset, the robustness of the threshold values for correlation coefficient and time analyzed needs to be rechecked."*

**Comment IV:**
Then, I would also suggest moving in this section the method used to define the rainfall events, using section 2.1, and 2.2 only to describe the dataset available.

**Reply:** We agree. Section "2.3 Method" was renamed to "2.3. Detection of Landslide Clusters and a new section "2.4 Rainfall Analysis" was created. 2.4 now contains all of the information previously in 2.2 describing how precipitation before landslide events was analyzed in order to better understand its impact:

*"2.4 Rainfall Analysis*
*In order to compare rainfall during a landslide event to overall rainfall at the location, the 95th percentile of precipitation excluding non-rainy days was determined for 10 years prior to the event. This comparison was also previously used by Kirschbaum et al (2015) to identify rainfall triggered landslide events. However, in their case, rainfall data from the Tropical Rainfall Measuring Mission (TRMM) was used for the time period 2000–2013 independent of the date of the landslide event. Due to its higher spatial resolution CHIRPS data was used here instead.*

*In addition to the 95th percentile of rainfall, the global rainfall threshold by Guzzetti et al. (2008) was also utilized to determine the likelihood of the individual landslide events being triggered by rainfall. In their study 2626 rainfall events that have resulted in shallow landslides and debris flows were analyzed in order to determine*

*the          following          global          rainfall          intensity–duration          threshold [http://rainfallthresholds.irpi.cnr.it]:*

$$I = 2.2 \cdot D^{-0.44} \qquad (1)$$

*Here the threshold intensity (I) was determined for each 24 hours starting with a duration (D) of 12 hours. This results in an average precipitation of 0.73 mm/h for D = 12 h, 0.45 mm/h for D = 36 h, and 0.35 mm/h for D = 60 h. The rainfall threshold was then compared to the cumulative mean precipitation of the rainfall event preceding each landslide event."*

**Comment V:**

Please consider using more tables in Section 3 to summarize and better describe the results obtained. Currently the text may result a bit confusing.

**Reply:** We agree. Thus, we added an additional row to Table 1 giving information on the global dataset and created Tables S2 and S3, providing information about trigger and size for each region. Because this information is also available in Figure 4, we decided to place these tables in the supplementary material.

**Table 1.** *Regional statistics for all landslide clusters (LC) with at least ten landslide events (LE).*

| Region | # LC | # LE | LE per LC | Average duration of LCs | LEs per day | Average area of LCs [km²] | Percentage of LE in a LC ≥ 10 LE |
|---|---|---|---|---|---|---|---|
| Global | 50 | 1,209 | 24.2 | 7 | 3.5 | 35,441 | 13 |
| West Coast, North America | 29 | 829 | 28.6 | 9 | 3.3 | 52,970 | 31 |
| Central and Eastern USA | 8 | 107 | 13.4 | 6 | 2.4 | 23,357 | 12 |
| South and Central America | 5 | 168 | 33.6 | 3 | 11.2 | 1,320 | 18 |
| Himalaya | 4 | 48 | 12.0 | 5 | 2.3 | 5,476 | 3 |
| South-East Asia | 4 | 57 | 14.3 | 5 | 3.2 | 5,143 | 4 |

**Table S2.** *Size of the landslides of clusters with at least 10 events for the different regions.*

| Region | Small | Medium | Large | Very large | Unknown |
|---|---|---|---|---|---|
| West Coast, North America | 26% | 29% | 1% | - | 43% |
| Central and Eastern USA | 62% | 38% | - | - | - |
| South and Central America | 4% | 88% | 5% | 3% | |
| Himalaya | 6% | 88% | 6% | - | - |
| South-East Asia | 4% | 88% | 9% | - | - |

*Table S3. Apparent trigger of the landslides of clusters with at least 10 events for the different regions.*

| Region | Downpour | Rain | Continuous rain | Snow melt | Flooding | Tropcial cycloon | Con-struction | Unknown |
|---|---|---|---|---|---|---|---|---|
| West Coast, North America | 27% | 19% | 3% | 2% | 1% | - | - | 47% |
| Central and Eastern USA | 64% | 15% | 9% | 3% | - | - | 1% | 6% |
| South and Central America | 85% | - | 8% | - | - | 7% | - | - |
| Himalaya | 40% | 4% | 48% | - | 2% | - | - | 6% |
| South-East Asia | 16% | 19% | - | - | - | 65% | - | - |

**Comment VI:**

Finally, I would suggest to clearly split the discussion from the results. Please consider creating Section 4. "Discussion. In this way the authors' comments are highlighted and easy to be understood for a reader.

**Reply:** We would like to keep our manuscript concise, thus we would like to keep our current structure.

**Specific comments**

**Comment VII:**

Avoid in the text: ">"," <", etc...

**Reply:** All Instances of ">" and "<" have been replaced with "more than"/"greater" and "less than" respectively.

**Comment VIII:**

Fig.1 difficult to distinguish among dots. I would suggest enlarging this figure.

**Reply:** We agree, the Figure was updated and now shows a heat map instead of individual dots.

[Figure]

*Figure 1. Heat map of all landslide events analyzed in this study and their size and apparent trigger. Overall a total of 9279 events were tested for clustering.*

**Comment IX:**

Please check the numbers for tables and figures (i.e., lines 122, 124: Fig. S1 ??; line 169: Table S1??)

**Reply:** With Table S1 and Fig. S1 (and following Figures S2 – S10) we meant to refer to tables in the supplementary material. This information is now given when first mentioning Figure S1 and Table S1:

"*A schematic drawing of this algorithm is provided in Fig. 2, and a more detailed flowchart in Fig. S1 in the supplementary material.*"
"*Table S1 in the supplementary material gives more detail of the 50 clusters with at least 10 events.*"